# Recent Advances in PROTAC-Based Antiviral Strategies

**DOI:** 10.3390/vaccines11020270

**Published:** 2023-01-27

**Authors:** Haleema Ahmad, Bushra Zia, Hashir Husain, Afzal Husain

**Affiliations:** 1Department of Biochemistry, Faculty of Life Sciences, Aligarh Muslim University, Aligarh 202002, India; 2Department of Biosciences, Faculty of Science, Integral University, Lucknow 226026, India

**Keywords:** PROTACs, targeted-protein degradation, viruses, antiviral, vaccine

## Abstract

Numerous mysteries of cell and molecular biology have been resolved through extensive research into intracellular processes, which has also resulted in the development of innovative technologies for the treatment of infectious and non-infectious diseases. Some of the deadliest diseases, accounting for a staggering number of deaths, have been caused by viruses. Conventional antiviral therapies have been unable to achieve a feat in combating viral infections. As a result, the healthcare system has come under tremendous pressure globally. Therefore, there is an urgent need to discover and develop newer therapeutic approaches against viruses. One such innovative approach that has recently garnered attention in the research world and can be exploited for developing antiviral therapeutic strategies is the PROteolysis TArgeting Chimeras (PROTAC) technology, in which heterobifunctional compounds are employed for the selective degradation of target proteins by the intracellular protein degradation machinery. This review covers the most recent advancements in PROTAC technology, its diversity and mode of action, and how it can be applied to open up new possibilities for creating cutting-edge antiviral treatments and vaccines.

## 1. Introduction

Investigation of intracellular processes has not only provided answers to various fundamental questions in cell and molecular biology but also led to the development of new technologies for the treatment of infectious and non-infectious diseases. For example, the research on understanding protein degradation and homeostasis has led to the development of technologies for targeted protein degradation, a new therapeutic modality currently being used to target biomedically relevant proteins that are not amenable to conventional small-molecule inhibitors. In targeted protein degradation, multi-specific small molecules redirect the cellular protein degradation machinery toward the target protein of interest (POI). The PROteolysis TArgeting Chimeras (PROTAC), a technology for targeted protein degradation, relies on using heterobifunctional molecules to recruit intracellular protein degradation machinery to the intracellular target protein of interest [1]. This chemically-induced proximity between protein degradation machinery and the target POI results in polyubiquitylation and proteasomal degradation of the target protein [2]. Despite the field of PROTAC technology being relatively new, PROTACs have found wide applications not just as a technical tool but also as a therapeutic approach for infectious and non-infectious diseases, including cancer and neurodegenerative diseases [3,4,5,6].

Viruses have been known to cause many global pandemics throughout the history of humankind, leading to massive deaths. Some of the deadliest diseases have been caused by viruses, including AIDS (acquired immunodeficiency syndrome), smallpox, Ebola, certain cancers, and the most recent pandemic caused by severe acute respiratory syndrome coronavirus 2 (SARS-CoV-2). This has put enormous pressure on the global healthcare system, which other diseases have already constrained. Some currently used antiviral strategies include small molecule inhibitors that inhibit the target viral proteins and vaccines that help generate a robust immune response against a viral pathogen. Small molecule inhibitors interact with the key proteins essential for viral replication at different stages of the viral life cycle. These conventional methods share limitations, including incomplete inhibition of viral proteins and poor immunogenic response. Extensive research has been conducted to determine viral targets for developing efficient antiviral therapies. Developing effective vaccines against viruses with a rapidly evolving genome is a huge challenge. The diversity in the genetic pool of such viruses makes it difficult to select and identify a viral epitope that can be used to elicit an effective immune response [7]. Existing strategies and technologies fail to decipher the evolving genome of viruses and thus lack specificity in targeting viruses. Newer technologies and approaches that help tackle the drawbacks of conventional vaccines are the need of the hour in the face of the ongoing pandemic.

This review discusses recent advancements in PROTAC technology and its potential to develop alternative antiviral therapeutic strategies. We aim to explore further the current understanding of this technology, its mechanism of action, diversity, and applicability as antiviral therapeutics, including its usage to create alternative vaccine strategies.

## 2. PROTAC as a Tool for Targeted Protein Degradation

The intracellular protein degradation involves an array of proteins, including chaperones and the ubiquitin-proteasome system [8]. While the ubiquitin-proteasome system removes the unfolded and damaged proteins, the chaperone corrects any misfolding of the proteins. The 26S ubiquitin-proteasome system consists of one or two 19S regulatory subunit(s) which dictate(s) substrate specificity for proteasome-dependent cleavage and a 20S proteasome core which degrades the unfolded proteins [9]. The proteins meant for degradation by the proteasome are modified post-translationally by covalent tagging with the small ubiquitin (Ub) protein. This ubiquitylation involves a cascade of three enzymes: an E1 ubiquitin-activating enzyme, which activates Ub in an ATP-dependent manner forming the E1-Ub conjugate, an E2 ubiquitin-conjugating enzyme, which catalyzes transesterification, resulting in the transfer of ubiquitin from E1 to E2, and finally, an E3 ubiquitin ligase, which binds both the E2-Ub and the protein substrate and transfers Ub from E2 to the lysine residues of the protein slated for degradation. Repetition of this reaction generates a polyubiquitin chain that directs the proteins for degradation by the 26S proteasome [10,11].

The concept of PROTAC was first reported in 2001 when Sakamoto et al. designed PROTAC to specifically degrade MetAP2 (methionine aminopeptidase-2) protein in Xenopus egg extract using a chimeric molecule comprising of MetAP2 ligand ovalicin linked to a phosphopeptide ligand of E3 ubiquitin ligase BTRC. Although peptide-based PROTACs efficiently degrade target proteins, there are some limitations that reduce their utility. For example, their labile nature, high molecular weight, and poor membrane permeability. These limitations were later overcome by synthetic small molecule PROTACs. The first small molecule PROTAC reported in 2008 was comprised of a selective androgen receptor modulator (SARM) as a ligand for the androgen receptor and nutlin as a ligand for the mouse double minute 2 homolog (MDM2) E3 ligase connected through a PEG-based linker [12]. Further improvements in the small molecule PROTACs expanded their use to in vivo depletion of the target proteins [13,14].

The fundamental structure of PROTAC consists of an “E3 moiety” that binds to the substrate binding domain of E3 ubiquitin ligase, a “protein targeting moiety” that binds to the POI and a linker that joins both the moieties (Figure 1). From recent studies, it has become clear that the effectiveness of overall degradation depends not only on the affinities of the “E3 moiety” and “protein targeting moiety” but also on the length and chemical makeup of the linker [15]. The length and chemical makeup play a crucial role in the formation of the ternary complex, target protein degradation efficiency, and target selectivity [16,17,18]. The substrate preferences of E3 ligases are influenced by the unique molecular architectures formed by the unique combinations of their scaffolding proteins, Ub-loaded E2, adaptor proteins, and substrate binding domains [10,18]. The human proteome currently contains more than 600 distinct E3 ligases.

PROTACs have quickly evolved and are considered a superior alternative to small molecule inhibitors [19,20]. The unique mechanism of PROTAC action enables rapid, long-lasting, and potent biological response against the target protein. The higher specificity of PROTACs reduces side effects that are associated with the off-target binding of conventional small molecule inhibitors [21]. PROTACs are also beneficial and preferred over small-molecule inhibitors as they can target multifunctional proteins with enzymatic and scaffolding roles [22,23].

## 3. Diversity of PROTACs

The shift in usage from peptide-based PROTACs to small molecule-based PROTACs has revolutionized the field of PROTACs. Small molecule-based PROTACs have been developed for the targeted protein degradation of various proteins such as ALK, Bcl-2 family proteins, Bcl-6, BCR-ABL, BRD4, BTK, FLT-3, HDAC family proteins, MDM2, PLK1, PRC2, and STAT3 among many others [24,25,26,27,28,29,30]. Only a handful of the 600 E3 ligases reported in the human proteome have been exploited for developing PROTACs [3,31]. PROTACs can be of many different types based on the number and type of E3 ligase used, the nature of protein targeting moiety, the number of targets degraded, and the type of interactions involved.

The four most commonly used PROTACs, based on the type of E3 ligases used, are MDM2-based, CRBN-based, IAP-based, and VHL based. MDM2 acts as an oncogene that suppresses the activity of the tumor suppressor P53 protein by inducing its ubiquitination and subsequent degradation by the proteasome [32]. The first MDM2-based PROTAC utilized Nutlin-3a as the MDM2 ligand and SARM as the androgen receptor ligand to degrade the androgen receptor. MDM2-based PROTACs have been used to degrade PARP1, BRD4, BTK, TRKC, and HSP90 [33,34,35,36,37].

Inhibitor of apoptosis proteins (IAPs) is a family of proteins that negatively regulate apoptosis and consist of eight members X1AP, cIAP1, cIAP2, Ts-IAP, KIAP, BIRC5, BRUCE, and NAIP. The IAP E3 ligase-based PROTACs against cellular retinoic acid binding proteins (CRABPI and CRABPII) utilized bestatin ester as an IAP-binding ligand and all-*trans* retinoic acid as ligands for CRABP I and CRABP II. Methyl bestatin and the IAP antagonist LCL161 were used to substitute bestatin to address the shortcomings of bestatin-based PROTACs, such as their low potencies and autoubiquitination of cIAP [14,38]. PROTACs that recruit IAP have been found subsequently target many other proteins, including BCL-xL and BCR-ABL, ER, AR, Retinoic Acid receptors, and transforming acidic coiled-coil-3-containing protein [39,40].

VHL (Von Hippel Ligand) is a ubiquitously expressed E3 ligase; hence, VHL-based PROTACs have found application in many cell types [41,42]. As the transcription factor HIF-α is the natural substrate of VHL, a seven amino acid peptide derived from HIF-α and dihydroxy-testosterone, a ligand androgen receptor (AR), was used to develop a VHL-based PROTAC to degrade AR [43]. The first small molecule, VHL-based PROTAC, was developed against estrogen-related receptor alpha and receptor-interacting serine/threonine kinase 2 [13]. 

Cereblon (CRBN) forms an E3 ubiquitin ligase complex with the damaged DNA binding protein 1 (DDB1), cullin-4A (CUL4A), and regulator of cullins 1 (ROC1). CRBN is a target of immunomodulatory drugs such as thalidomide, lenalidomide and pomalidomide [44]. In contrast to VHL, CRBN-based PROTACs have been shown to degrade a broader range of proteins [41]. CRBN-based PROTACs are not as tissue-selective as VHL-based PROTACs as they are ubiquitously expressed. In addition, they have a low molecular weight, making them feasible to be developed as an orally bioavailable PROTAC.

Based on the number of E3 ligases used, PROTACs are classified into homobivalent and trivalent PROTACs. Homobivalent PROTACs involve a ternary complex of either a pair of similar kinds of E3 ligases or different kinds of E3 ligases joined by a suitable linker. The first kind of homobivalent PROTACs employs two similar E3 ligases and causes the dimer thus formed to self-ubiquitinate the E3 ligase involved [45]. Some of the commonly used PROTACs of this category include VHL-VHL-based and CRBN-CRBN-based homobivalent PROTACs. [46] The second kind involves two different types of E3 ligases and thus leads to the degradation of either one type of ligase or both [47,48]. Examples of this category include CRBN-VHL-based and MDM2-CRBN-based PROTACs [49,50].

Trivalent PROTACs are also known as two-headed PROTACs due to their degradative actions on two different target proteins. This dual activity permits the trivalent PROTAC molecule to simultaneously target two different proteins of interest [51]. Li and his co-workers synthesized the first trivalent PROTAC molecule using the dual targeting properties of the trivalent PROTACs to target two completely different proteins, PARP1 and EGFR [52]. They were, however, limited in their action by their large molecular weight, poor cellular permeability, and lower solubility. 

The covalent PROTACs, yet another class of PROTACs, form an even more stable ternary complex providing improved target specificity and enhanced binding affinities. The covalent PROTACs are designed with the formation of a covalent interaction of a linker with a target protein or an E3 ligase, and these interactions may be reversible or irreversible [53]. These PROTACs have been reported to have high potency and selectivity coupled with covalent interactions that are in a simultaneous loop of formation and dissolution after the target protein has been selected and degraded [54]. 

Moreover, some other types are developed with a slight divergence in design from the conventional small molecule-based PROTACs. Instead of using small molecules as linkers between an E3 ligase and the protein of interest, BioPROTACs are developed by reengineering the substrate recognition domain of the E3 ligase such that the domain is replaced by a peptide that directly binds to the protein of interest. This eliminates the necessity of finding appropriate ligands for both the E3 ligase and the protein of interest. Though BioPROTACs exhibit poor permeability, they are highly target-specific. 

Another category includes AbTACs or antibody-based PROTACs, which exploit bispecific antibodies to cause the targeted degradation of cell surface proteins. A bispecific antibody combines two different proteins and labels them for degradation [55]. These mimic the structure of a PROTAC molecule. Also, highly specific AbTACs can be created by exploiting recombinant technology, which will enhance the affinity of the synthesized molecule for the protein of interest [56]. 

## 4. PROTAC-Based Antiviral Strategies

Viruses caused massive deaths during the 20th century; for instance, smallpox killed up to 400 million people [57], and influenza caused around 100 million deaths during the great Spanish flu outbreak in 1918–1919 [58]. Since it was initially identified in 1981, the human immunodeficiency virus (HIV) epidemic has been responsible for 35 million fatalities [59,60]. As of 25 January 2022, confirmed cases of COVID-19 had reached 352 million, with 5.60 million deaths (https://covid19.who.int, accessed on 28 November 2022). This pandemic was caused by the SARS-CoV-2 and is referred to as “among the deadliest pandemics of the past century” [61,62]. Viral infections have become a serious concern to public health and safety. Currently, the prevention and treatment of human viral infections mainly rely on a combination of drugs and vaccines [63,64]. However, growing drug-resistant strains pose a challenge for currently available antiviral therapeutic strategies, while vaccinations frequently fail to protect against altered or novel viruses [65]. Therefore, it is crucial to find cutting-edge antiviral therapeutic strategies based on either novel drug targets or innovative targeting or vaccination strategies. As discussed above, PROTAC technology has been widely investigated for the targeted protein degradation of POI in the context of cancer. But, of late, many advances have been made in understanding the role of PROTACs as antiviral therapeutics. Various PROTAC-based antiviral therapeutic strategies with improved resistance profiles have been recently explored. By putting these novel approaches into practice, it may be possible to develop powerful antiviral therapeutics capable of combating the current and future risks posed by emerging and re-emerging viral pandemics. 

### 4.1. PROTAC Virus, a Novel Vaccine Strategy

A vaccine is a biological preparation of attenuated or killed disease-causing microorganisms that has the potential to elicit an immune response. Vaccines may also be formulated using toxins released by such microorganisms or their surface receptors that mimic their identity in the host body. Vaccines induce an immune response upon detecting a foreign entity unfamiliar to the body and hence help develop humoral immunity. They are of various types: attenuated, inactivated, toxoid, subunit, conjugate, heterotypic, and genetic [66]. Conventionally, live attenuated vaccines have been employed to prevent influenza infections. But they are limited by suboptimal immunogenicity, safety concerns, and cumbersome manufacturing processes [67]. 

Recently, Si et al. described a novel PROTAC-based approach to generate engineered attenuated influenza virus strains for vaccination purposes. They employ PROTAC technology for targeted protein degradation of selected viral proteins of the influenza virus by the host ubiquitin-proteasome system, dramatically attenuating viral replication. Although PROTAC viruses were defective in their replication, they could elicit robust immune responses. To generate attenuated influenza PROTAC viruses, they engineered the genome of influenza A by linking the proteasome targeting domain (PTD) to the matrix gene segments. The PTD, a heptapeptide with a sequence ALAPYIP, is recognized by VHL E3 ubiquitin ligase. The tagging of viral proteins by PTD results in their polyubiquitylation by VHL E3 ligase, followed by proteolysis by the host ubiquitin-proteasome system. As PTD peptide is linked to the viral protein through a tobacco etch virus (TEV) cleavage site linker peptide (ENLYFQG), it can be conditionally cleaved in cell lines that stably express TEV protease, sparing the proteolysis of the PTD-tagged target proteins. Therefore, influenza virus replication was not defective in cell lines stably expressing TEV, producing attenuated virus particles for vaccine manufacture.

Si. et al. tested eight different influenza A proteins (M1, PB2, PB1, PA, NP, M2, NEP, and NS1) for PTD-tagging dependent proteolysis and attenuation of replication in conventional Madin-Darby canine kidney 2 (MDCK2) cells and PTD-cleavage efficiency and protection from proteolysis in TEV expressing MDCK2 cells (MDCK-TEVp). Although PTD-tagging led to efficient proteolysis of each of the eight proteins in MDCK2 cells, the PTD-cleavage efficiency and, thus, the protection from proteolysis in MDCK-TEVp was highly variable. This was largely attributed to the accessibility of the TEV-cleavage site of the PTD-tagged proteins. Among the eight proteins, PTD-tagging of M1 (M1-PTD) showed efficient proteolysis of M1 protein in MDCK2 cells, resulting in >20,000-fold decrement in replication competence compared to wild-type virus. On the contrary, the M1-PTD virus efficiently replicated in TEV expressing MDCK-TEVp cell line (Figure 1). Also, the M1-PTD virus was not only highly attenuated in BALB/c mice and ferrets but was also genetically stable. The group also tested the ability of M1-PTD to induce an immune response in mice and ferrets and detected titters for HI (Hemagglutinin Inhibition), NT (Neutralization), HA (hemagglutinin), and internally conserved nucleoprotein antibodies significantly higher than those immunized with inactivated influenza vaccine (IIV) or by cold-adapted influenza vaccine (CAIV). A robust and adept T-cell immune response was also detected due to the enhanced presentation of degraded viral peptide antigens by MHC molecules.

PROTAC viruses hold the potential as an ideal vaccine candidate. An ideal vaccine is capable of attaining a level of sufficient attenuation in the host for safety while also retaining robust immunogenicity in cell lines [68,69]. In contrast to the conventional methodologies of vaccine production, PROTAC exploits the degraded viral peptides generated from the proteasomal degradation pathway to trigger an efficient immune response [70]. PROTAC technology is more efficient than other methodologies of attenuation due to improved safety and a cumulative loss in efficacy or productivity. Another prominent challenge to the conventional attenuation-based approaches is the immune escape due to rapid viral evolution. Hence, PROTAC technology has emerged as a prominent option for generating safer and more effective vaccines.

### 4.2. Proteases-Targeting PROTAC

The viruses encode one or more proteases as a typical technique to support replication with a compacted genome. To produce mature viral proteins, the viral genome encodes a polyprotein with an integrated viral protease that cleaves the polyprotein at several particular places. As a result of their necessity for reproduction, viral proteases are excellent therapeutic targets. The viral life cycle and replication depend on the viral proteases. Proteases from various viral families differ from one another in terms of structure, catalytic mechanism, and preferred substrate. Viral proteases have a specific substrate preference that can be used when designing inhibitors to create strong and selective drug-like compounds. Extensive studies on the role of viral proteases hint towards their important role in the replication phase of viruses. Hence, viral proteases are a common target for inhibiting viral replication in antiviral strategies. Various conventional small molecule inhibitors against viral proteases have been developed over the last few decades. In order to overcome the drug resistance as well as improve the efficacy of the anti-protease small molecule inhibitors, some researchers applied PROTAC-based target protein degradation of viral proteases. 

A PROTAC compound against a non-structural 3/4A (NS3/4A) serine protease of the hepatitis C virus (HCV) was described [71]. Telaprevir, a reversible covalent inhibitor that binds to the active site of the HCV NS3/4A serine protease, was used to develop a PROTAC degrader for this protease (Figure 2). Telaprevir’s crystal structure with the viral protease revealed that its pyrazine ring is solvent exposed, allowing it to be derivatized with various linkers conjugated to ligands of CRBN, the substrate receptor for the CUL4-RBX1-DDB1-CRBN E3 ubiquitin ligase. The resulting telaprevir-based bivalent degrader, DGY-08-097, caused both inhibition and selective degradation of the HCV NS3/4A protease in cellular infection models. Additional PROTAC molecules based on boceprevir, a synthetic tripeptide that selectively inhibits NS3/4A protease, can be developed as a potential antiviral against HCV.

Since the propagation of SARS-CoV-2 began in 2019, there has been an urgent need for drug or vaccine development against it. More than 22 million people were affected by SARS-CoV-2 worldwide (www.coranavirus.jdu.edu, accessed on 28 November 2022). SARS-CoV-2 are members of the Beta coronavirus genus’ Coronaviridae family. At 5′ of the SARS-CoV-2 genome, there are two overlapping ORFs: ORF1a and ORF1b. ORF1a and ORF1b encode polyprotein 1a and 1b (pp1a and pp1b), which are cleaved into 16 NSPs by the protease activity of two cysteine proteases: PL^pro^ (nsp3) and M^pro^ (nsp5) [72]. Therefore, developing PROTACs against proteases could be a potential target. Most proposed inhibitors are peptidomimetics and tiny compounds with Michael acceptors like aldehydes and ketones. These inhibitors share the property of a P1-lactam group acting as a glutamine mimetic. Boceprevir, a direct-acting antiviral (DAA), stops the spread of viruses. Boceprevir, an orally available synthetic tripeptide that inhibits the non-structural protein 3 and 4A complex (NS3/NS4A), could also be used to develop a potent PROTAC-based antiviral. A single-digit IC50 value was obtained for boceprevir, an FDA-approved HCV NS3/4A protease inhibitor, to inhibit SARS-CoV-2 M^pro^ [73]. The rapid identification of SARS-CoV-2 M^pro^ inhibitors as clinical candidates, which outpaces that of other viral protease therapeutic targets, emphasizes the significance of understanding the structure and mechanism of viral proteases. A review by Liu et al. also hypothesized that inhibition of Cov 3C-like protease through PROTAC degraders could be the basis for next-generation anti-CoV drugs [74]. These novel antiviral targets could circumvent viral variation and address drug resistance to conventional enzymatic inhibitors. The discovery supported the idea that these small-molecule degraders could prevent or cure viral variations linked to resistance to conventional inhibitors since they were less susceptible to changes impacting ligand binding. 

### 4.3. Surface Receptor-Targeting PROTACs

The viral envelope is a lipid bilayer that encapsulates the capsid. It is embedded with many glycoproteins on its surface, enabling the virus to attach to the host cell’s receptors. On the other hand, many coronaviruses (CoVs) possess unique envelope and membrane proteins essential for the sustenance of viral replication, apart from these glycoproteins present in the envelope. The viral surface proteins, like hemagglutinin and neuraminidase, are emerging as the new target for antiviral therapies due to their stark difference from the receptors on the host’s cell surface. Hemagglutinin is a lectin involved in the virus’s attachment to its receptor on the cell surface by recognizing terminal sialic acid residues present on them [75]. Neuraminidase helps move the virus from one sialic acid residue to another by cleaving the bond between hemagglutinin present on the viral envelope and sialic acid residues on the host cell membrane [76]. The cycle continues, and the virus traverses the host’s cell membrane until it recognizes and attaches to the proper cell receptor for its entry into the host cell. Similarly, the activity of neuraminidase also makes it possible for freshly created virions to leave the surface of the host cell. Any damage to the viral envelope may irreversibly damage it due to the generation of reduced oxygen species because the viruses lack the machinery to repair them [77,78].

Oseltamivir, a neuraminidase inhibitor, prevents the exit of the newly synthesized viral particles from the host cell. Neuraminidase is involved in cleaving the bond of hemagglutinin with sialic acid residues that attach the newly synthesized virions to the host cell’s surface. It has been widely used in the treatment of infections caused by influenza A and influenza B viruses. Oseltamivir-based PROTACs against the influenza virus were developed by connecting E3 ligase ligands on the N-terminal and carboxyl-terminal of Oseltamivir via diversified linkers (Figure 2). Hydrophobic interactions and hydrogen bonds of the synthesized PROTAC molecules with neuraminidase and VHL ligase were involved in the formation of a stable ternary complex. The N-substituted oseltamivir-based PROTACs were more effective than their carboxylate counterparts [79]. The mechanism of action of oseltamivir-based PROTACs involves the targeting and degradation of neuraminidase via the ubiquitin-proteasome pathway. The oseltamivir-based PROTACs tend to work dually. Firstly, these PROTACs have oseltamivir connected to one end, which has a strong affinity for neuraminidase and inhibits its action. Secondly, these PROTACs employ the E3 ligases to degrade the neuraminidase enzyme. Inhibition of the neuraminidase activity and its subsequently degradation ensures that the virions synthesized in the host cell do not leave and remain attached to the host cell. These PROTACs were effective against oseltamivir-resistant strains on further evaluation [79].

Apart from targeting the most commonly present glycoproteins in the envelope of viruses, certain other envelope and membrane proteins can also be targeted for degradation that can subsequently hinder viral productivity, as in the case of coronaviruses. Coronaviruses (CoVs) are enveloped viruses containing a positive, single-stranded RNA genome packaged within a capsid. The capsid consists of the nucleocapsid protein N, which is further surrounded by a membrane that contains three proteins: the membrane protein (M) and the envelope protein (E), which are involved in the virus budding process, and the spike glycoprotein (S), which is a key player in binding the host receptor and mediating membrane fusion and virus entry into host cells [80,81]. The envelope protein E within the SARS-CoV-2 can be a potential target protein of interest. Many factors make this protein feasible as a target protein. The protein E is the least abundant among the other proteins and is the only protein that is not glycosylated [82]. As there is a lack of glycosylation, the protein can be easily accessible to small molecules to engage with it. Since its absence or inactivation can directly alter viral assembly, membrane permeabilizing activity, and other parameters, targeting envelope protein E affects virulence. Additionally, viruses frequently associate with receptor proteins on cell surfaces to enter human cells; for instance, the SARS-CoV-2 interacts with the angiotensin-converting enzyme 2 (ACE2) receptor [83]. 

PROTAC technology may be an effective method to degrade these proteins and subsequently stop the virus for those targets that are hard to discover direct inhibitors for or non-druggable targets, such as Nsp1, Nsp3b, Nsp3c, E-channel, etc. The drugs having the highest potential binding affinities for the SARS-CoV-2 proteins are gramicidin S and tyrocidine A. Through ACE-2 receptors, S-glycoprotein is crucial for coronavirus attachment to the host cell surface. The association between S-glycoprotein and ACE-2 was broken in binding experiments with Dactinomycin and Gramicidin S to S-glycoprotein, breaking the link between viral S-glycoprotein and the host’s ACE-2 receptor [84]. These molecules could be used to design potential PROTAC molecules to target surface proteins. For the S protein, only one compound, natural hesperidin, was found to target the binding between the S receptor binding domain (RBD) and human ACE2 [85]. Any small molecule bound to S may interfere with S’s re-folding, inhibiting the viral infection process.

Furthermore, small molecules targeting any part of the S protein may be a good starting point for designing PROTAC-based therapy. Aside from the S protein, the E protein (E-channel) performs crucial biological tasks for the coronavirus’s structural integrity and host pathogenicity. For N proteins in host cells to bind with coronavirus RNA effectively, they require the NRBD and CRBD of the coronavirus N protein. Therefore, the NRBD and CRBD domains of the E protein or the N protein can be exploited as targets for developing antiviral medications. The PROTAC against protein E promotes proteasomal degradation. The antiviral PROTAC also has the additional advantage of providing an antiviral immune response. The innate immune response against the viral infection will promote its clearance through its presentation by MHC-I, its proteasomal degradation and development of T-cell antibodies. An increased presentation of viral proteins through MHC-I may lead to increased T-cell activity against the viral protein. 

### 4.4. Host Protein-Targeting PROTAC

After the infection of the host cell with the virus, the viruses hijack the host cell’s machinery to synthesize viral proteins and nucleic acids for their rapid multiplication and assembly. Several enzymes are necessary for viral replication and productivity. These include polymerases, endonucleases, and ligases involved in various nucleic acid transactions. A new shift in the development of antiviral therapeutics has been seen in seeking host proteins as targets. It will enable tackling the resurgence of drug resistance in viruses. Furthermore, since viral pathogens require host machinery to replicate their genetic material, targeting host proteins successfully will help bypass the hurdle of frequently tweaking antivirals frequently to effectively target the rapidly mutating viral genome.

Drug repurposing was the most sought-after strategy to tackle the COVID-19 pandemic by identifying existing drugs as effective antivirals. One such drug was indomethacin (INM), which was discovered as a potential antiviral drug candidate using a network-based deep-learning methodology and omics-based analysis. INM is a non-steroid anti-inflammatory drug with analgesic and antipyretic properties and is known to inhibit PGES-2 (prostaglandin E synthase type-2), an enzyme known to be involved in eicosanoid biosynthesis [86]. It was shown that INM inhibits the SARS-CoV-2 replication cycle by activating the protein kinase R pathway, inhibiting protein synthesis in virus-infected cells. INM inhibits the host enzyme PGES-2, which interacts with non-structural protein (NSP7), an essential component of the viral primase complex along with NSP8. The complex is a part of the viral RNA polymerase machinery and is conserved in many variants of CoVs [87,88]. Hence, targeted protein degradation of PGES-2 by PROTACs may act as a universal anti-CoV drug.

INM-based PROTACs were synthesized by the conjugation of INM with the VHL E3 ligase ligand through an aliphatic or polyethylene glycol linker [89]. Since INM-based PROTACs target the host protein (PGES-2), their efficacy as antiviral therapeutics is eminent in overcoming drug resistance issues, usually due to the resurgence of variations in the genome of viral proteins. PROTACs based on INM have been developed as an effective antiviral therapy that inhibits not only the enzymatic activity of the proteins but also their scaffolding activity, downregulating the subsequent cascade of reactions that could otherwise revive viral synthesis despite inhibiting the viral proteins required for replication (Figure 3).

Similarly, it is well established that cyclin-dependent kinases (CDKs) play an important role in the life cycles of not only DNA and RNA viruses [90]. Consistently, the inhibitors of CDK1, CDK2, CDK3, CDK4/6, CDK5, CDK7, and CDK9 show potent antiviral activities against a variety of viruses, including HSV, HIV, Human cytomegalovirus (HCMV), and SARS-CoV-2 [90]. These findings prompted Hahn et al. to develop THAL-SNS032, a commercially available CDK9-directed PROTAC (Figure 3). This PROTAC is produced by coupling the E3-recruiting unit thalidomide to the CDK inhibitor SNS032. THAL-SNS032 mostly binds and hence causes degradation of CDK9, but also CDKs 1, 2, and 7 [91]. In addition, THAL-SNS032 also targets the HCMV-encoded ortholog of CDKs, pUL97, which play important roles in viral replication. HCMV is a significant opportunistic human pathogen widespread worldwide, with seroprevalence ranging from 40% to 95% in the adult population. It was discovered that THAL-SNS032 was sensitive to the HCMV virus, whereas other viruses exhibited insensitive behavior.

Even though a vast majority of approved antiviral drugs target viral proteins, they tend to pose the risk of the emergence of drug-resistant viral strains, limiting the scope of developing efficient pan-viral therapies. This is because some antiviral therapies may tend to work against a particular strain but will not work against another strain with the same target sites due to the introduction of mutations. To counter these, host-directed therapies are currently being explored. Viruses rely heavily on the host machinery to ensure their replication, and targeting these key host targets will assist in hindering the replication cycle of viruses. Additionally, these host targets can be utilized to develop broad-spectrum antiviral medicines because they have a very limited mutation potential in contrast to rapidly evolving viral genomes. Introducing PROTAC technology in the targeted degradation of host proteins will significantly enhance the specificity of the antiviral strategies. Hence, many new host targets can be explored for PROTACs, including the host receptor, CCR5 (chemokine receptor type 5) in the case of HIV type 1, and tight junction proteins like Claudin1 in the case of HCV.

### 4.5. Miscellaneous

The Hepatitis B virus (HBV) is known to infect more than a third of the world’s population, and its infection is a major risk factor for the development of hepatocellular carcinoma. The viral infection and productivity are maintained by the X-protein of the virus, which is involved in the transactivation and dysregulation of multiple cancer-associated genes, DNA repair mechanisms, and cell apoptosis [92,93]. A novel PROTAC has been designed that is capable of simultaneously degrading and inhibiting the function of protein X [94]. The N-terminal oligomerization domain was fused to the C-terminal instability domain of the protein X to construct the novel PROTAC. Additionally, it was made cell-permeable by adding a polyarginine cell-penetrating peptide. The X-protein would bind to the oligomerization domain, and the instability domain of the X-protein would tag the protein for proteasomal degradation. All in all, cell-penetrating PROTACs based on the oligomerization and instability domains of the X-protein can antagonize the X-protein and play an important role in causing its elimination. These novel PROTACs can be employed not only as therapeutics in the treatment of Hepatitis B virus infection but also in the prevention of hepatocellular carcinoma [94].

Human immunodeficiency virus (HIV) is a retrovirus that causes acquired immunodeficiency syndrome (AIDS) [95]. It is also known as human T-cell leukemia (lymphotropic) virus type III (HTLV-III). The majority of HIV viral proteins are made up of accessory proteins, virus-specific enzymes, and structural proteins. All fully functioning viruses have the same well-studied viral host protein interactions, essential for their activity. There are three HIV-related enzymes: reverse transcriptase, protease, and integrase. The multidomain enzyme HIV-1 integrase is necessary to integrate viral DNA into the host genome. Three domains make up the enzyme. The His2Cys2 motif in the N-terminal domain chelates zinc, the catalytic DDE motif in the core domain is necessary for the enzyme’s enzymatic activity, and the SH3-like fold in the C-terminal domain unspecifically binds DNA [96]. By causing the breakdown of viral proteins, the host ubiquitin-proteasome pathway (UPP) performs crucial functions in host defense against viruses. HIV-1 Human Interaction Database (available at https://www.ncbi.nlm.nih.gov/genome/viruses/retroviruses/hiv-1/interactions/, accessed on 30 November 2022) is a comprehensive database of HIV-1 human protein interactions that enables the identification of regions involved in virus-host protein interactions that can be used for designing chimeric E3 ligases. Substrates for ubiquitin-proteasome degradation have been successfully established. In the study by Zhang et al., they attempted to design and construct artificial chimeric ubiquitin ligases (E3s) based on known human E3s in order to target HIV-1 integrase for ubiquitin-proteasome pathway-mediated degradation manually. It was designed in a way that a functional chimeric E3 that can target HIV-1 integrase, the substrate-binding domains of various natural E3s were replaced with the HIV-1 integrase binding domain (IBD) of human LEDGF/P75 protein or the enzymatic domains of E3s were linked with the IBD directly and succeeded in creating chimeric E3 146LIS that was capable of targeting HIV-1 NL4-3 integrase for Lys48-specific polyubiquitination and degradation by 26S proteasome [97]. The table below categorizes the numerous viral targets that have been targeted for degradation using PROTAC-based antiviral methods (Table 1).

## 5. Advantages of PROTAC-Based Antiviral Strategies

PROTAC technology has found profound applicability in developing antiviral therapeutics due to its ability to generate a sustained and potent biological response during the targeted degradation of essential viral proteins. As PROTACs are highly target-specific, they result in the highly selective degradation of target proteins of either the viral pathogen or the host. Due to their higher efficiency and selectivity, PROTACs work at lower dosing regimens than conventional small molecule inhibitors to produce a therapeutic response and thus reduce the side effects. Advancements in PROTAC technology have led to dramatic improvements in the efficacy and potency of PROTACs. The development of small molecule-based PROTACs overcame the drawbacks of peptide-based PROTACs, such as their poor cell penetration ability, high molecular weight, labile nature of peptide bonds, and sometimes low potency. Some of these small molecules based PROTACs exhibit high potency even in nanomolar concentrations [98].

Generally, PROTACs target both the enzymatic and scaffolding activities of multifunctional proteins by efficiently and specifically degrading the targeted proteins. Target specificity of PROTACs is ensured by the protein–protein interactions induced by the formation of the ternary complex, which converts the ‘promiscuous’ ligands to selective degraders [42]. Although developing a potent and efficient inhibitor against a specific isoform of a protein is difficult, PROTAC-mediated degradation helps dictate the different degradation outcomes of the different ternary complexes formed by each isoform. This enables enhanced isoform selectivity and helps in eliminating undesirable off-target activity [99]. Therefore, PROTACs may be employed in diverse activities, including selective degradation of a single isoform and pan-isoform catalytic inhibition [31]. For instance, palbociclib-based PROTAC induces selective degradation of only CDK6 but not CDK4, despite palbociclib being a potent inhibitor of both CDK6 and CDK4 [99].

Mutations in the target proteins make them resistant to small molecule inhibitors [22]. However, PROTACs can overcome drug resistance resulting from mutations in a target POI or small molecule-induced compensatory increase in the level of the target proteins [24]. This is because the efficiency of a PROTAC molecule in inducing targeted protein degradation is dependent more on the formation of a stable ternary complex comprising of the E3 ligase and the target protein than their respective binding affinities. Hence, these protein-protein interactions generated due to the formation of the ternary complex will help in stabilization despite the weak binding affinity between the PROTAC molecule and the target protein. The selectivity of the PROTACs can be further improved by adjusting the length and composition of linkers, altering the binding of ligands to the target proteins or E3 ligases, and using different E3 ligases [22,29,100,101,102]. PROTACs are also more efficient than small-molecule inhibitors because they strongly affect the downstream signaling pathways. This sustains the inhibition for a longer time compared to the small molecule protein inhibitors [22]. Studies on FAK, FLT3-ITD, BCR-ABL, and PARP have already confirmed these advantages of PROTACs over small molecule inhibitors. [22,103,104].

PROTACs may play multiple roles while acting as antiviral therapeutics. PROTACs cause the degradation of the viral target protein by polyubiquitylation and subsequent proteasomal degradation, followed by the generation of the immune response against the degraded viral peptides. The proteasomal degradation of the viral target peptides causes the MHC class I molecules to load them onto their binding grooves and present the viral epitopes to the T-cells for subsequent activation [105]. These lead to the generation of antibodies against the target viral proteins. In addition, enhanced polyubiquitination of the viral proteins leads to enhanced presentation of viral peptides by MHC Class I [106].

The evolving genome of a virus makes it challenging for a particular antiviral therapy to function against it for a longer time but also limits the duration of immunity elicited by infection and vaccination. Studies suggest that PROTAC technology can increase the efficacy of existing antiviral therapeutics [79]. Differently recruited E3 ligases in the ternary complex enable the selection of different target viral proteins with variable specificities. Similarly, unlike small molecules that inhibit a single subunit, PROTACs induce the degradation of multiple proteins in a complex [16].

Over the last two decades, the development of antiviral PROTACs was limited to targeting viral or host proteins. However, the recent application of PROTAC technology to generate live attenuated viral vaccine strains could potentially streamline the cumbersome vaccine manufacturing process. Utilizing PROTAC technology to create live attenuated viral vaccine strains has several benefits [107]. This approach attenuates viral replication to a very low level in experiments compared to other approaches that have safety concerns. The PROTAC-based vaccination approach utilizing live attenuated virus has the potential to elicit an effective immune response against not just one but multiple circulating seasonal or pandemic viral strains by providing a sufficient antigenic match between the PROTAC-attenuated viral vaccine strain and the target virus. This is in contrast to the existing vaccines, such as CA1V and FluMist, wherein only the envelope proteins are chosen from the circulating strains while the remaining six internal genes are from a cold-adapted master donor virus. The internally conserved viral proteins trigger long-term cellular immunity [1]. A PROTAC-based vaccine is expected to be a highly cost-effective technology with high specificity and potency in comparison to other conventional methodologies. It requires a shorter duration as compared to existing approaches [108].

The fast evasion of viruses caused by low-fidelity polymerases, many offspring viruses emerging from a single infected cell, and quick replication cycles can be combated by PROTACs that provide their unique resistance-reducing properties. Additionally, the potential of PROTACs is not limited to enzymes but may also be used for non-enzymatic target proteins, i.e., new PROTACs may be created by identifying small molecule ligands binding viral proteins to increase the spectrum of virus-encoded targets that are drug-accessible. PROTACs may be produced to increase the range of drug-accessible virus-encoded targets. Key viral replication cycle phases that were previously unavailable may now be taken advantage of through the PROTAC-mediated degradation of viral proteins with either regulatory or structural roles. The capacity of PROTACs to degrade their targets in several consecutive cycles (sub-stoichiometric mode of action) is crucial since it is not just limited to the primary target but may also include binding partners in the context of multiprotein complexes (Table 2). As a result, highly organized collections of viral structures made up of many copies of structural proteins from the virus might be sensitive to drug-targeting.

## 6. Limitations

As discussed above, PROTACs are advantageous over small molecule inhibitors [109]. However, some concerns linked to PROTAC-based antiviral therapeutics should be addressed before their therapeutic use. Although PROTACs are more powerful than other small molecule inhibitors and completely deplete the desired target protein in the cell, they might cause on-target toxicity in the host cell and may impair normal cell function. Some POIs targeted by PROTACs can have both enzymatic as well as other scaffold functions that may be important for normal cellular functioning. Thus their total elimination can be toxic to the cell [110]. The degradation of the target protein of interest by the PROTAC may also degrade proteins that are directly or indirectly associated with the target protein, resulting in off-target effects. Off-target effects can also cause neomorphic interactions if the PROTAC binds to neo-substrates [111]. Therefore, future research should focus on strategies that can further reduce both on-target and off-target toxicities associated with PROTACs.

## 7. Future Implications

PROTAC technology is expected to result in the development of promising and potentially next-generation antivirals in the future. The PROTAC technology, unlike the traditional antivirals, has an expanded function against a repertoire of pathogenic proteins and can be effective against resistance acquired by mutating and adapting viruses. The technology in question can be further expanded by exploring a greater number of E3 ligases, apart from the existing collection of known ligases, as genomic alterations in the core of the E3 ligases’ machinery have flared up the resistance to therapies based on PROTAC technology. Developing the appropriate ligands for the target protein and E3 ligase can be assisted by artificial intelligence (protein structure prediction), virtual drug screening, and DEL screening (DNA-encoded library screening). For the quick and efficient construction of large-scale libraries of PROTAC molecules, it is necessary to optimize screened ligands and create effective synthesis methods based on high throughput screening.

Some of the existing drawbacks of PROTAC technology include its large molecular weight, cellular permeability, and metabolic instability of the small molecule-based PROTACs. CLIPTAC (CLIck-formed Proteolysis targeting chimera) technology has been used to counter this issue [112]. In click chemistry, the bifunctional molecule is divided into two parts: a ligand for the target protein, and the other part that binds to the ligase. The parts assemble to form a PROTAC molecule via a rapid reaction in the cell. Because both parts have a low molecular weight and improved cellular permeability, they significantly improve the technology. It has been shown that linkers also significantly impact the activity, selectivity, and druggability of PROTAC compounds. Shorter linker lengths may disrupt the formation of a stable ternary complex due to steric hindrance and restrict the PROTAC molecule from degrading the target protein. In contrast, a longer linker length may increase the molecular weight, reducing cellular permeability and efficacy. Increasing the rigidity of the linker enhances oral bioavailability by constraining a PROTAC molecule in its bioactive conformation.

Despite the increasing surge in the development of PROTAC technology, several concerns are associated with it. Although PROTAC technology is a promising tool for targeting the “undruggable proteome,” only a handful of these sites have been targeted by PROTACs thus far. Additionally, the pharmacodynamics and pharmacokinetics of these molecules need to be effectively analyzed and evaluated because the traditional methodologies typically used to evaluate the kinetics and dynamics of conventional small molecules that follow a stoichiometric mechanism of action are ineffective. There is also a growing need to screen for more protein ligands that effectively target protein-protein interactions. Furthermore, since PROTACs can completely degrade the target protein, there is a high probability of cytopathic effects arising from the complete elimination of otherwise essential proteins.

Better molecular design of PROTACs will be possible in the future thanks to more complex information gathered by X-ray crystallography or cryo-electron microscopy. Recent advances in alphafold2′s capacity to predict proteins and their associated complex structures may impact how PROTACs are designed. Studies have been carried out to discover newer targets that can effectively enhance the specificity of the technology. One such example is Ribonuclease Targeting Chimeras or RIBOTACs, which have emerged as a new class of small molecules that can potentially target diverse RNA types [113]. These RNA-degrading molecules can selectively bind an RNA-binding molecule to a latent ribonuclease, RNase L, which is involved in the degradation of the RNA molecule upon its activation [114]. RIBOTACs contain a tetra-adenylate component, similar to an oligoadenylate, that causes the dimerization and activation of RNase L, leading to viral RNA degradation.

Emerging and re-emerging viral infectious diseases pose a persistent pandemic threat to humankind, not only because of their negative impact on public health but also because of the possible worldwide economic, social, and political repercussions. Vaccination is the only sought after that could be a preventive measure, and the molecular-based approach may provide a more diverse defense against newly emerging viruses than conventional ones. Further extensive studies that elaborate on the possibility of newer targets and enable enhanced target specificity can contribute towards the advancement in PROTAC technology as potent antiviral therapeutics in the coming years.

## Figures and Tables

**Figure 1 vaccines-11-00270-f001:**
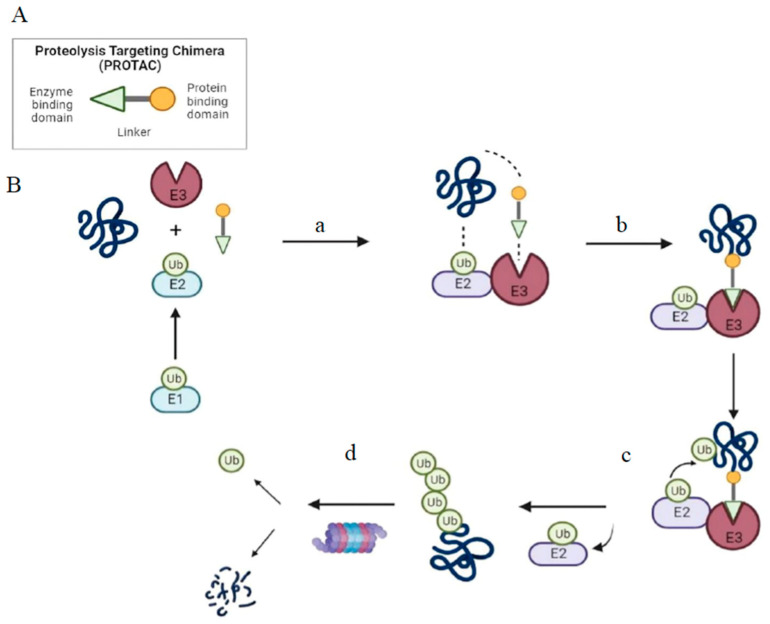
Structure and mechanism of PROTAC-based degradation. (**A**) General structure of a PROTAC: The E3 ligase targeting “anchor” (green) is connected to the specific POI targeting warhead (yellow) via a variable linker; (**B**) Mechanism of PROTAC-mediated target degradation (a) Ub transfer from E1 to E2, which is followed by complex formation with an E3 ligase; (b) the PROTAC binds to both the E3 ligase and POI to form a TC. This brings the E2 ligase into proximity to the POI; (c) this leads to the transfer of multiple Ub units to surface-exposed lysine residues; (d) the resulting polyubiquitin chain is recognized by the proteasome, leading to the proteolytic degradation of the POI. Ub, Ubiquitin; POI, Protein of interest; TC, Ternary complex.

**Figure 2 vaccines-11-00270-f002:**
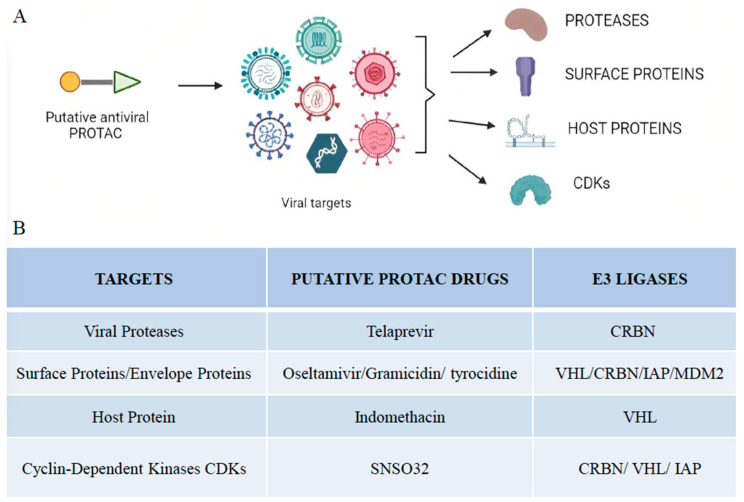
Viral targets of different PROTAC-based approaches. (**A**) Illustration of putative antiviral PROTACs targeting different viral targets, including proteases, surface proteins, host proteins, and cyclin-dependent kinases. (**B**) Summary of viral targets and their putative antiviral drugs and E3 ligases that make up the PROTAC molecule.

**Figure 3 vaccines-11-00270-f003:**
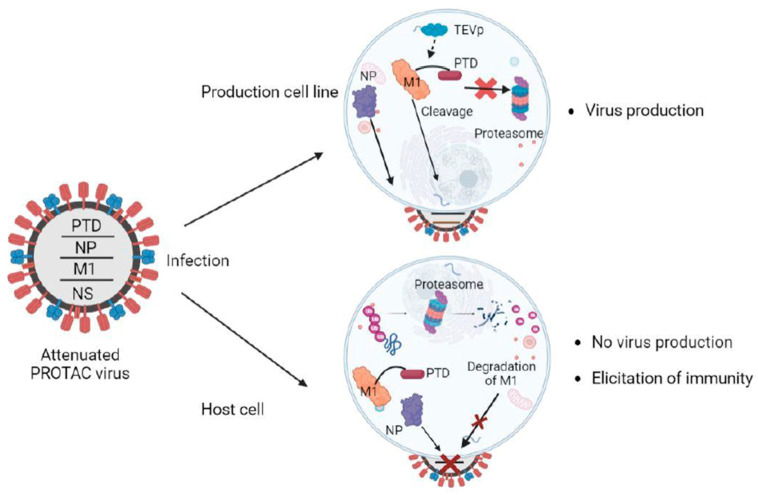
Overview of the PROTAC-based vaccine design. The ubiquitinated viral protein (M1-PTD) of PROTAC viruses is targeted and degraded by the proteasome in host cells, attenuating their reproduction and causing a deficit in protein synthesis. The influenza nucleoprotein (NP), which is not fused to PTD, is not degraded. In the production cell line, cells are designed to produce the TEV protease (TEVp), which cleaves the PTD, and the PROTAC viruses proliferate well for vaccine production. M1, matrix protein; Ub, ubiquitin; PTD, proteasome-targeting domain.

**Table 1 vaccines-11-00270-t001:** Summary of PROTACs as an antiviral strategy.

Target	Examples	E3 Ligase	Mechanism of Action	Ref.
**Viral proteases**				
	Telaprevir	CRBN	Telaprevir acts as a protein ligand in the ternary complex and may reversibly bind to and inhibit the viral proteases.	[71]
**Surface proteins**				
HA	Oseltamivir	VHL/CRBN	Oseltamivir binds to the neuraminidase enzyme in the viral envelope and inhibits it. The PROTAC molecule employs the cellular ligases for the subsequent degradation of neuraminidase.	[79]
2. NA				
3. Envelope protein E	Thalidomide	VHL	Targets envelop protein E and lead to its degradation via the proteasomal degradation pathway.	
**Host protein**				
PGES-2	Indomethacin		Indomethacin binds to the host protein, PGES-2. PGES2 and NSP-7 form the viral primase complex. Targeting PGES-2 for degradation hinders the formation of the viral polymerase. Alternatively, it interacts with NSPs in the viral primase complex and regulates the PKR pathway to inhibit protein synthesis.	[89]
2. CDK’s	SNS032	CRBN/IAP/VHL	CDK-directed PROTACs cause the degradation of other essential CDKs that participate in the formation of the nuclear capsid of the virus.	[91]

**Table 2 vaccines-11-00270-t002:** Summary of advantages of PROTACs over other conventional antivirals.

Target	PROTACs	Conventional Antivirals
**Specificity**	PROTACs offer highly specific and precise machinery to degrade the target protein and eliminate its activity.	They may not be highly specific and are also unable to cause complete elimination of the target proteins because they can only inhibit the activity of the protein.
**Efficiency**	They are highly potent and do not depend on the target protein’s high affinity. The formation of a ternary complex is sufficient to induce protein degradation.	They require a very high binding affinity with the target protein and hence are not very efficient in action.
**Mechanism of action**	They exhibit a catalytic mechanism of action because a single molecule can degrade more than one target protein molecule.	They have been known to function in a stoichiometric manner; that is, a single molecule can only inhibit a single molecule of the target protein.
**Dosage**	Since they are highly specific, potent, and efficient, even a nanomolar concentration range can induce targeted protein degradation.	Due to the stoichiometric mechanism of action and weaker efficiency, they are required to be administered in higher doses.
**Resistance**	It is highly unlikely for viruses to develop resistance to PROTACs.	Prolonged exposure can induce drug resistance because of antigenic shifts and drift in the viral genome.

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
