# Peer review of "Recent Advances in PROTAC-Based Antiviral Strategies"

_vaccines, 2023, doi:10.3390/vaccines11020270_

Round 1
Reviewer 1 Report
PROTAC-based vaccines are an interesting choice to target certain viruses.
PROTACs are compounds (heterobifunctional in nature), where half of the molecule is intended to target the protein of interest, whereas the other half binds to E3 ligase activity enzymes.
Furthermore, these strategies have recently gained more importance because of their event-driven pharmacology than the typical occupancy-driven pharmacology. Additionally, the going clinical trials of PROTAC (ARV-based PROTACs), showed their true potential to use for direct human use.
Unfortunately, like any other drug targeting strategy, PROTAC does suffer some of the typical flaws, which makes less favorable choices against many diseases.
However, implementing the PROTAC strategy as an antiviral requires a robust understanding of the chemical biology of a particular virus.
There are certain points that authors require to address in the paper.
1. Make a table categorizing these antiviral PROTACs strategies, to enhance the readability of the paper.
2. As an expert, the authors require to add a critical perspective on the limitations of these PROTAC strategies and, point out the possible improvements that can be implemented in the near future.
Author Response
Reviewer 1:
PROTAC-based vaccines are an interesting choice to target certain viruses.
PROTACs are compounds (heterobifunctional in nature), where half of the molecule is intended to target the protein of interest, whereas the other half binds to E3 ligase activity enzymes.
Furthermore, these strategies have recently gained more importance because of their event-driven pharmacology than the typical occupancy-driven pharmacology. Additionally, the going clinical trials of PROTAC (ARV-based PROTACs), showed their true potential to use for direct human use.
Unfortunately, like any other drug targeting strategy, PROTAC does suffer some of the typical flaws, which makes less favorable choices against many diseases.
However, implementing the PROTAC strategy as an antiviral requires a robust understanding of the chemical biology of a particular virus.
There are certain points that authors require to address in the paper.
Thank you for your valuable comments and suggestions. We strongly believe that these comments will help improve this review. Please see our point-to-point response to your comments below.
- Make a table categorizing these antiviral PROTACs strategies, to enhance the readability of the paper.
Different strategies for antiviral PROTAC are included in the manuscript as Fig 2. However, as per the suggestion by the reviewer, Table 1 consisting of different antiviral strategy and their mechanism of action has been incorporated in the revised manuscript on page number 13.
- As an expert, the authors require to add a critical perspective on the limitations of these PROTAC strategies and, point out the possible improvements that can be implemented in the near future.
Limitations of PROTAC strategies and its improvement were included in the manuscript in the advantages and future direction section of the manuscript however as asked by the reviewer, we have incorporated a subheading on limitations and the improvements of the PROTAC technology as an antiviral strategy in the revised manuscript at page numbers 15 and 17.
Reviewer 2 Report
This review article highlights the recent advancements in protein degradation technology (PROTAC) and how PROTACs can be leveraged for creating antiviral treatments and vaccines.
1. First, this is a relevant article to the Vaccines journal. This review summarizes how PROTAC was discovered, its evolution and future directions in the context of vaccines and antiviral technology.
2. The authors logically articulated their thoughts with appropriate references to PROTACs. The language is concise and understandable.
3. Overall content in this work is highly informative. It speaks volumes about the good side and caveats of PROTAC technology.
4. Since there are multiple review articles in the PROTAC space and fewer articles in the context of using PROTACs for antiviral and/or vaccine technology, I wish more emphasis was placed on the latter. This is only a minor concern.
5. I would request the authors to re-work Figure no 1. The quality of the figure is not to the mark. It is pixelated. Figure 2 has minor spelling corrections. Also, please fix the alignment issues in the paragraphs that come after the figures.
6. I would suggest a table on how PROTAC-based antivirals and vaccines are better than conventional drugs. Kindly speculate on the chances of the virus becoming resistant to PROTACs.
I'm happy with the scientific content of the paper and I recommend this work for publication with minor changes. Congratulations to the team.
Author Response
Reviewer 2
This review article highlights the recent advancements in protein degradation technology (PROTAC) and how PROTACs can be leveraged for creating antiviral treatments and vaccines.
1. First, this is a relevant article to the Vaccines journal. This review summarizes how PROTAC was discovered, its evolution, and future directions in the context of vaccines and antiviral technology.
2. The authors logically articulated their thoughts with appropriate references to PROTACs. The language is concise and understandable.
3. Overall content in this work is highly informative. It speaks volumes about the good side and caveats of PROTAC technology.
4. Since there are multiple review articles in the PROTAC space and fewer articles in the context of using PROTACs for antiviral and/or vaccine technology, I wish more emphasis was placed on the latter. This is only a minor concern.
Thank you for reviewing our manuscript. Your comments have helped us make this manuscript more complete and apt. As per your suggestion, few points have been added in the revised manuscript at page number 15.
- I would request the authors to re-work Figure no 1. The quality of the figure is not to the mark. It is pixelated. Figure 2 has minor spelling corrections. Also, please fix the alignment issues in the paragraphs that come after the figures.
Thank you for pointing this out. We have now provided the image with better resolution and have also fixed the alignment issues.
- I would suggest a table on how PROTAC-based antivirals and vaccines are better than conventional drugs. Kindly speculate on the chances of the virus becoming resistant to PROTACs.
Thank you for the suggestion. As per your suggestion, Table 2, has been incorporated in the revised manuscript at page 16.
I'm happy with the scientific content of the paper and I recommend this work for publication with minor changes. Congratulations to the team.
Thank you
Reviewer 3 Report
In general, the article is written competently and interestingly, makes a good impression. However, part of the data was not included in the review, they should be added for completeness of coverage of this topic:
1- Use of PROTAC technology against hepatitis B virus https://www.sciencedirect.com/science/article/pii/S0006291X14017926?via%3Dihub
2- For cytomegalovirus https://www.mdpi.com/1422-0067/22/23/12858/htm 3- For HIV https://www.ncbi.nlm.nih.gov/pmc/articles/PMC8257815/ and https://grantome.com/grant/NIH/R01-AI084140-03
It is also recommended to move the section "PROTAC-based antiviral strategies" to the introduction.
Author Response
Reviewer 3:
In general, the article is written competently and interestingly, making a good impression. However, part of the data was not included in the review, they should be added for completeness of coverage of this topic:
Thank you for reviewing our manuscript. We think by adding these points, the manuscript looks better and complete. As per your suggestion, we have added a sub heading at page number 12 and included the text.
- Use of PROTAC technology against hepatitis B virus https://www.sciencedirect.com/science/article/pii/S0006291X14017926?via%3Dihub
- For cytomegalovirus https://www.mdpi.com/1422-0067/22/23/12858/htm
As per your suggestion, we found that the text relevant to the topic was present in the manuscript however the reference was missing, than you for pointing it out, the reference has been cited in the text at page number 11.
- For HIV https://www.ncbi.nlm.nih.gov/pmc/articles/PMC8257815/ and https://grantome.com/grant/NIH/R01-AI084140-03
It is also recommended to move the section "PROTAC-based antiviral strategies" to the introduction.
In the introduction section, we have only briefly introduced the PROTAC technology in general and its applications as an antiviral therapeutic strategy. However, in the section "PROTAC-based antiviral strategies," the focus is explicitly shifted toward viruses. Therefore, we think it would be better to keep this section separate and not as part of the introduction.
Round 2
Reviewer 3 Report
The article can be published.